METHODS AND RESOURCES

# Knowledge-guided analysis of "omics" data using the KnowEnG cloud platform

**Charles Blatti, III**[1], **Amin Emad**[1,2], **Matthew J. Berry**[3], **Lisa Gatzke**[3], **Milt Epstein**[1], **Daniel Lanier**[1], **Pramod Rizal**[3], **Jing Ge**[1], **Xiaoxia Liao**[3], **Omar Sobh**[1], **Mike Lambert**[3], **Corey S. Post**[4], **Jinfeng Xiao**[4], **Peter Groves**[3], **Aidan T. Epstein**[1], **Xi Chen**[1], **Subhashini Srinivasan**[1], **Erik Lehnert**[5], **Krishna R. Kalari**[6], **Liewei Wang**[7], **Richard M. Weinshilboum**[7], **Jun S. Song**[1,8,9], **C. Victor Jongeneel**[1], **Jiawei Han**[4,9], **Umberto Ravaioli**[10], **Nahil Sobh**[1‡], **Colleen B. Bushell**[3‡], **Saurabh Sinha**[1,4,9‡]*

**1** Carl R. Woese Institute for Genomic Biology, University of Illinois at Urbana-Champaign, Urbana, Illinois, United States of America, **2** Department of Electrical and Computer Engineering, McGill University, Montreal, Canada, **3** National Center for Supercomputing Applications, University of Illinois at Urbana-Champaign, Urbana, Illinois, United States of America, **4** Department of Computer Science, University of Illinois at Urbana-Champaign, Urbana, Illinois, United States of America, **5** Seven Bridges Genomics, Charlestown, Massachusetts, United States of America, **6** Department of Health Sciences Research, Mayo Clinic, Rochester, Minnesota, United States of America, **7** Department of Molecular Pharmacology and Experimental Therapeutics, Mayo Clinic, Rochester, Minnesota, United States of America, **8** Department of Physics, University of Illinois at Urbana-Champaign, Urbana, Illinois, United States of America, **9** Cancer Center at Illinois, University of Illinois at Urbana-Champaign, Urbana, Illinois, United States of America, **10** Department of Electrical and Computer Engineering, University of Illinois at Urbana-Champaign, Urbana, Illinois, United States of America

☯ These authors contributed equally to this work.
‡These authors jointly supervised this work.
* sinhas@illinois.edu

**Data Availability Statement:** Data for the primary analyses are available in GitHub at https://github.com/KnowEnG/quickstart-demos/tree/master/publication_data/blatti_et_al_2019 and in the Supporting Information files.

## Abstract

We present Knowledge Engine for Genomics (KnowEnG), a free-to-use computational system for analysis of genomics data sets, designed to accelerate biomedical discovery. It includes tools for popular bioinformatics tasks such as gene prioritization, sample clustering, gene set analysis, and expression signature analysis. The system specializes in "knowledge-guided" data mining and machine learning algorithms, in which user-provided data are analyzed in light of prior information about genes, aggregated from numerous knowledge bases and encoded in a massive "Knowledge Network." KnowEnG adheres to "FAIR" principles (findable, accessible, interoperable, and reuseable): its tools are easily portable to diverse computing environments, run on the cloud for scalable and cost-effective execution, and are interoperable with other computing platforms. The analysis tools are made available through multiple access modes, including a web portal with specialized visualization modules. We demonstrate the KnowEnG system's potential value in democratization of advanced tools for the modern genomics era through several case studies that use its tools to recreate and expand upon the published analysis of cancer data sets.

**Funding:** This effort was part of KnowEng BD2K Center supported by grant U54GM114838 awarded by National Institute of General Medical Sciences through funds provided by the trans-National Institutes of Health Big Data to Knowledge (BD2K) initiative (https://commonfund.nih.gov/bd2k). It was also funded in part by the Cancer Center at Illinois. The funders had no role in study design, data collection and analysis, decision to publish, or preparation of the manuscript.

**Competing interests:** The authors have declared that no competing interests exist.

**Abbreviations:** AML, acute myeloid leukemia; ARI, adjusted Rand index; AWS, Amazon Web Services; BRCA, breast cancer; CDH1, cadherin 1; COCA, cluster of cluster assignment; CTNNB1, catenin beta 1; CWL, common workflow language; EGFR, epidermal growth factor receptor; EMT, epithelial to mesenchymal transition; ESCC, esophageal squamous cell carcinoma; ESR1, estrogen receptor 1; FAIR, findable, accessible, interoperable, and reusable; FOXA1, forkhead box A1; FOXM1, forkhead box M1; GATA3, GATA binding protein 3; GBM, glioblastoma; GDC, Genomic Data Commons; GO, Gene Ontology; HNSCC, head and neck squamous cell carcinoma; KnowEnG, Knowledge Engine for Genomics; KRAS, Kirsten rat sarcoma viral oncogene homolog; LUSC, lung squamous cell carcinoma; miRNA, microRNA; NBS, network-based stratification; NRAS, neuroblastoma RAS viral oncogene homolog; NRF2, nuclear factor erythroid 2-related factor 2; PAM50, Prediction Analysis of Microarray 50; RNA-seq, RNA sequencing; RWR, random walk with restart; SB-CGC, Seven Bridges Cancer Genomics Cloud; TCGA, The Cancer Genome Atlas.

# Introduction

The rapid growth of genomics data sets [1] and efforts to consolidate diverse data sets into common portals [2] have created an urgent need today for software frameworks that can be easily applied to these genomic "big data" to extract biological and medical insights from them [3]. Here, we present "KnowEnG" (Knowledge Engine for Genomics, pronounced "know-ing"), a cloud-based engine that provides a suite of powerful and easy-to-use machine learning tools for analysis of genomics data sets. These tools, also referred to as "pipelines," perform common bioinformatics analyses such as clustering of samples, gene prioritization, gene set characterization, and signature analysis. The tools are geared toward diverse omics data sets that can be represented as spreadsheets or tables (genes x samples) that record typical genomic profiles, such as gene expression, mutation counts, etc., for a collection of samples, at the resolution of individual genes. The pipelines help identify biologically meaningful patterns in the provided spreadsheet data, through ab initio analysis as well as by contextualizing with prior knowledge. The utility of KnowEnG is increased by co-localization of its tools with prior knowledge data sets from a large variety of sources.

## Diverse computing environments for KnowEnG

The genomics computing infrastructure of the future has to be adapted to the diverse ecosystem of data sets and tools that will continue to flourish in genomic research. In particular, tools must be "findable, accessible, interoperable, and reusable" [4], i.e., comply with the "FAIR" principles that guide the modern vision of biological data science. In recognition of these principles, software components of the KnowEnG system are packaged using state-of-the-art technology [5] that makes them highly portable and amenable to scalable execution in varying computing environments. A convenient way to access the system is through a web portal that links to a KnowEnG server (https://knoweng.org/analyze/; also see Appendix A in S1 File) running on Amazon Web Services (AWS). A user can upload their genomics data set as a spreadsheet and then execute available pipelines (Fig 1A and 1B and Appendix B in S1 File). Often, the resulting outputs of one KnowEnG pipeline can be further analyzed using another pipeline, and the system facilitates such "handover" between pipelines. For added security and control, users may also create a personal instance of the KnowEnG server and web portal using their AWS accounts (Appendix C in S1 File). This design feature can help meet challenges of heavy computing loads faced by a public analytics server. Computationally savvy users may invoke the pipelines and avail of additional functionalities through Jupyter [6] notebooks (https://knowtebook.knoweng.org) from a dedicated KnowEnG server. A third mode of access, created for cancer researchers, is via the Cancer Genomics Cloud resource built by Seven Bridges (SB-CGC) [7], in which users may directly access large cancer data sets, such as those generated by the Cancer Genome Atlas (TCGA) program [8], and analyze them using KnowEnG pipelines (https://cgc.sbgenomics.com/public/apps#q?search=knoweng) without transferring the data from AWS. Through these varied access modes, KnowEnG facilitates accessibility, interoperability, and reusability of its tools, marking a significant step towards realizing the "FAIR" vision.

## Knowledge Network–guided analysis

An important feature of KnowEnG pipelines is that they can incorporate large-scale prior knowledge about genes into analyses of the user's data set. A basic form of such "knowledge-guided" analysis is already common, in which the researcher performs statistical analysis of an experimental data set and then interprets the results in the light of prior knowledge from publicly available gene annotation repositories such as Gene Ontology (GO) [9], Reactome [10],

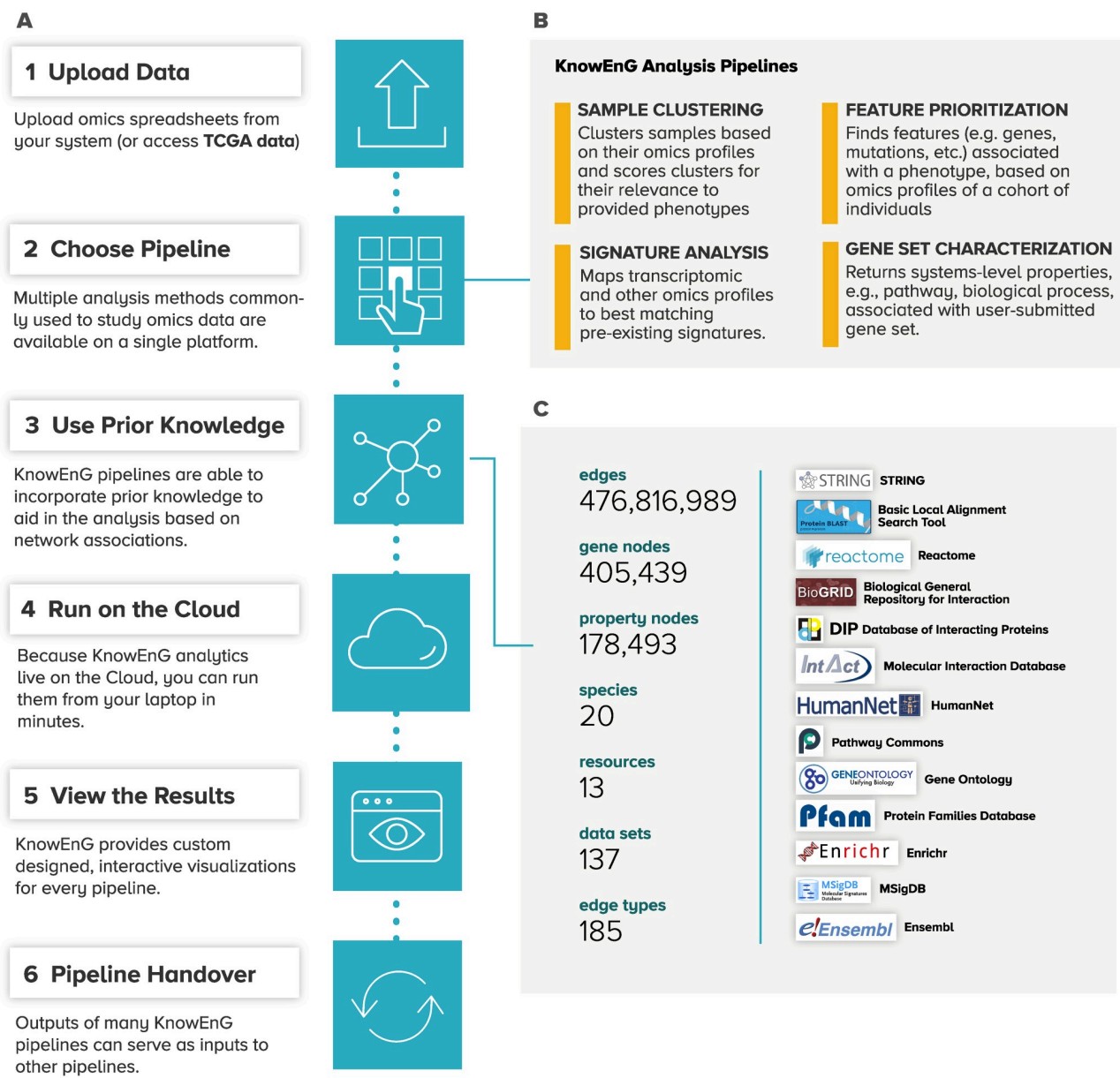

**Fig 1. Overview of KnowEnG platform.** (A) Typical workflow steps for KnowEnG: genomic analysis on the cloud. (B) Analytical functionalities are organized as "pipelines" for common tasks such as clustering, gene prioritization, gene set analysis, and signature analysis. Each pipeline offers various options to customize the analysis, including use of prior knowledge. (C) The KnowEnG Knowledge Network represents prior knowledge that may be used during analysis. Nodes in the network represent genes and biological properties, and edges represent either annotations of gene properties or gene-gene relationships. Network contents are summarized on the left with logos of data sources shown on the right. KnowEnG, Knowledge Engine for Genomics; TCGA, the Cancer Genome Atlas.

etc. KnowEnG makes this analytic process more rigorous by adapting its statistical tools to be directly guided by the vast data in such public repositories of gene annotations and interactions. In doing so, KnowEnG builds on a rich tradition of knowledge-guided analysis methods that have been previously reported for a variety of biological research tasks including (1) clustering of samples into cancer subtypes [11–14], (2) finding markers and drivers of disease [15–20], (3) prediction of patient survival [21,22] or cancer metastases [23], (4) characterization of experimental gene sets [24–28], and (5) prediction of gene functions [29–31]. KnowEnG also

breaks the logistical barriers associated with utilizing large databases of prior knowledge, by co-locating its "knowledge-guided analysis" tools with a diverse knowledgebase compiled from numerous popular repositories. The knowledgebase is encoded in a massive heterogeneous network called the "Knowledge Network," whose nodes are genes/proteins and whose edges represent properties (e.g., pathway membership) and mutual relationships (e.g., protein-protein interaction) of the nodes (Fig 1C). The network represents annotations of 41 different types from 20 species and 13 different data sources and includes 476 million edges, 405,000 gene nodes, and 178,000 property nodes; the network is regularly updated via a "one-click" internal system (Appendix A in S2 File). Users typically select the annotation type that is most relevant for guiding their analysis (Appendix D in S1 File) in the course of launching a pipeline. The Knowledge Network is also available as a stand-alone resource that allows subnetworks associated with a knowledge type to be retrieved (Appendix E in S1 File).

Here, we demonstrate the main functionalities, features, and interfaces of KnowEnG in the context of 2 influential data sets in cancer genomics [32,33]. We reproduce several key analyses of the original cancer studies in the KnowEnG system to highlight the ease-of-use with which multiple analysis pipelines can be invoked to generate publishable general insights and extract specific hypotheses from the data. We also present novel knowledge-guided analyses on these data sets that often result in more significant findings and provide a multifaceted narrative of the insights that the data have to offer. The scope of KnowEnG analytics goes far beyond cancer analysis, however, with the system currently supporting analysis of users' genomics data from any of 20 model organisms. We explain later (see Discussion) how its tools are broadly applicable to genomics data sets from any biological domain in any of the supported species.

## Results

### Overview of 3 "case studies"

We begin with an overview of the case studies used as demonstrations of KnowEnG pipelines. These pipelines analyze spreadsheets of genes (rows) by samples (columns), which tabulate numeric data on each gene in each sample condition. The data may come from a variety of sources, e.g., high-throughput transcriptomics assays using various technologies, mutation counts at the gene level, copy number variations, etc. These data are then analyzed through one of the KnowEnG pipelines, depending on the biomedical questions of interest, and the results are visualized in the platform and optionally passed on to additional pipelines for deeper investigation (Fig 2A). Additional information and simple instructions for easily reproducing the main analyses presented in this paper on the KnowEnG web server are found in Appendix F in S1 File.

In Case Study 1 (Fig 2B), we analyzed multiomics data on patients from a pan-cancer study [32]. We first used the "Sample Clustering" pipeline on somatic mutation data to identify clusters of patients with similar mutation profiles. We used knowledge-guided clustering in this step and found it to reveal an improved grouping of patients compared to traditional clustering, as judged by survival characteristics of the resulting clusters. We used the "Gene Prioritization" pipeline to identify genes that are most associated with each patient group and characterized the pathways related to those genes using the "Gene Set Characterization" pipeline. We also grouped the pan-cancer cohort of patients using other omics data and combined the results to obtain a single clustering based on all available data types, which we passed to the "Spreadsheet Visualizer" module to visually explore different aspects of the results including the improved significance of these clusters with overall patient survival.

In Case Study 2 (Fig 2C), we used the knowledge-guided Gene Prioritization pipeline to discover genes whose expression (from RNA sequencing [RNA-seq]) associated with each tumor

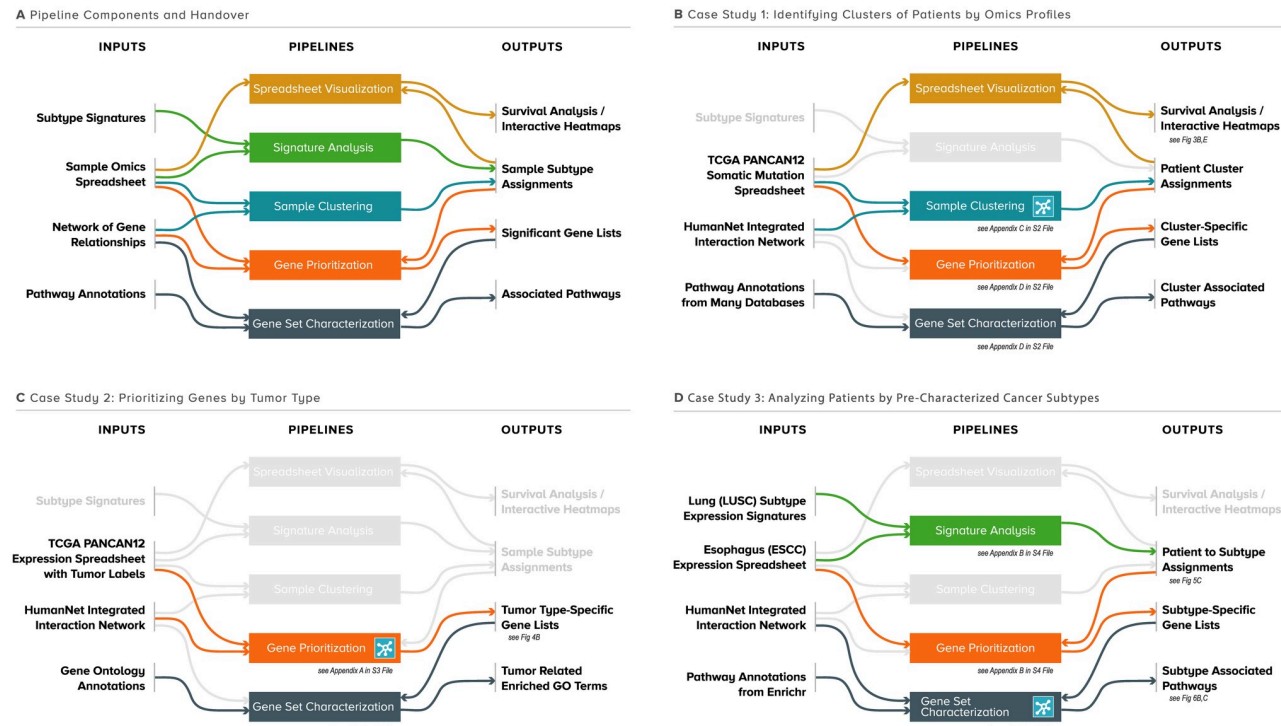

**Fig 2. Case studies demonstrating knowledge-guided analysis.** (A) Each of the KnowEnG analysis pipelines is shown as a box with corresponding colored arrows connecting to their general inputs and outputs. Because the outputs of a pipeline are often valid inputs for another pipeline, a convenient "handover" mechanism in the KnowEnG web portal enables chaining together multiple pipelines, facilitating deeper and multifaceted analysis of the user's data. (B) The analysis workflow for Case Study 1 with unused pipelines shaded gray. Simple descriptions of the case study–specific inputs and outputs are provided, along with notes on where to find the detailed methods or figures of the results. The pipeline step of the workflow that incorporates knowledge-guided analysis is indicated with the blue network icon. (C) Analysis workflow for Case Study 2. (D) Analysis workflow for Case Study 3. ESCC, esophageal squamous cell carcinoma; GO, Gene Ontology; KnowEnG, Knowledge Engine for Genomics; LUSC, lung squamous cell carcinoma; TCGA, the Cancer Genome Atlas.

type (as defined by tissue of origin) in the pan-cancer data set and found the discovered genes to include known drivers of the respective cancer types. We used the Gene Set Characterization pipeline to perform enrichment analysis of these prioritized genes and identify pathways and biological processes associated with each cancer type. We also found the prioritized genes to form a strong signature for the cancer types, in the sense that patient groups identified by examining expression levels of those genes were as informative of survival as groups identified from transcriptome-wide profiles.

In Case Study 3 (Fig 2D), we reproduced key findings of a transcriptomic study of esophageal squamous cell carcinoma (ESCC) [33] using the "Signature Analysis" pipeline. We assigned subtypes to ESCC patient transcriptomic profiles by relying on previously known signatures of lung squamous cell carcinoma (LUSC) subtypes, which provide clues into prognosis and chemotherapeutic resistance. We then identified genes and pathways associated with each ESCC subtype. Our main goal in this case study was to demonstrate how KnowEnG functionalities can be easily accessed on a third-party platform (Seven Bridges Cancer Genomics Cloud) where the data set resides. This flexibility allows researchers to seamlessly combine KnowEnG tools with other specialized tools that are available on the Seven Bridges platform when undertaking more complex projects with many different analytical components.

In the following sections, we describe each of the above case studies in detail. Our goal is to present typical usage of KnowEnG pipelines through examples taken from impactful published

studies, rather than as user manuals, which are also available through the KnowEnG website in help text and videos (Table A in S1 Data). These case studies illustrate how the pipelines can lead to broad insights from omics data sets and also more specific hypotheses such as genes and pathways involved in a process can emerge from follow-up analyses, also within KnowEnG.

## Case Study 1: Clustering of pan-cancer data

As a first demonstration of the analytic capabilities of KnowEnG, we describe how the Sample Clustering pipeline can be used to group genomic profiles in a knowledge-guided manner. Clustering is one of the most widely used tools in bioinformatics [34] and can help identify subgroups of samples that represent distinct biological or pathological states [35]; patient stratification in cancer, where subtypes are defined based on molecular markers [36], is a prime example. The same clustering tools are often applied to different types of genomic profiles, including gene expression, mutation counts, copy number mutations, etc. [32]. However, clustering of somatic mutation profiles of cancer patients presents a significant obstacle, because each profile is sparse (a minuscule fraction of genomic loci are mutated) and has little direct similarity to other profiles. As an example of a data set that presents this challenge, we worked with somatic mutation profiles of 3,276 tumor samples spanning 12 cancer types (Appendix B in S2 File) from the "pancan12" data set generated by the TCGA consortium [32]. (This large data set provides a natural "ground truth," viz., tumor type, for assessing clustering methods). We first used the "standard" mode of KnowEnG's Sample Clustering pipeline, viz., Hierarchical Clustering, in 6 different algorithmic configurations to identify 14 clusters (so as to match that in the original publication [32]) of tumor samples based on their somatic mutations. (The standard mode of this pipeline also offers K-means clustering). This failed to produce meaningful clusters, and almost every clustering result exhibited strong "resolution bias" [37], with one cluster comprising over 90% of the samples (Appendix C S2 File and Table E in S2 Data). The sole exception was clustering with Jaccard similarity and complete linkage [38], and even here the largest cluster had over 70% of the samples; we will refer to this below as the standard clustering. This initial analysis illustrates the challenge in clustering somatic mutation profiles: because of their high dimensionality and sparsity, biologically related profiles often do not harbor shared mutations and are not grouped together [11], ultimately leading to many small and one or few large clusters.

**Knowledge-guided clustering of mutation profiles.** Knowledge-guided clustering powered by the Knowledge Network offers a possible solution to the problem just noted. Here, prior knowledge of gene-gene relationships encoded in the network is used to recognize when somatic mutations in different genes may be functionally related, thus allowing more subtle forms of similarity between mutation profiles to be exploited in grouping patients. The knowledge-guided option of the Sample Clustering pipeline (Fig 3A) implements the "Network-based Stratification" (NBS) algorithm of Hofree and colleagues [11], in which a random-walk method makes patient mutation profiles less sparse by borrowing information from the Knowledge Network before the actual clustering step. We used knowledge-guided clustering with the HumanNet Integrated network (hnInt) [39] as prior knowledge to group patients into 14 clusters. This yielded more size-balanced clusters; the largest cluster included 30% of the 3,276 patients. To test whether patient groups identified from mutation profiles are tied to their phenotypic characteristics, we performed Kaplan-Meier survival analysis (Fig 3B). A log-rank test revealed highly significant distinction across the clusters in terms of survival probabilities ($p = 3.7 \times 10^{-33}$), which was clearly better than that observed in the standard clustering ($p = 7.4 \times 10^{-10}$; Figure F in S2 File). Notably, the original clustering analysis of mutation

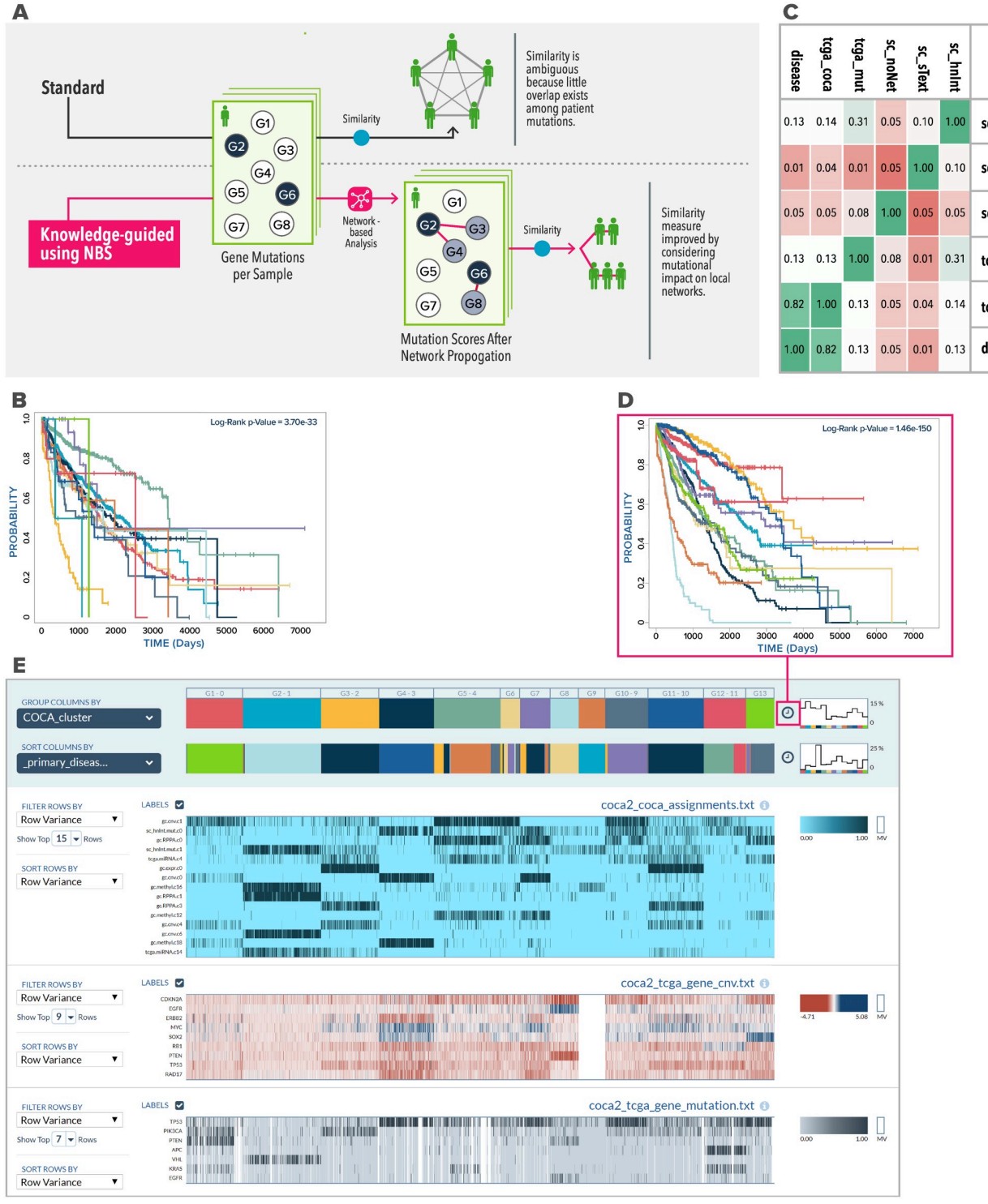

**Fig 3. Knowledge-guided sample clustering.** (A) Knowledge-guided sample clustering, illustrated in the context of somatic mutation profiles of cancer patients. Because mutations are rare, 2 patients may not have mutations to the same gene(s), and their mutual similarity will be modest. In the knowledge-guided mode (bottom), similarities between patient profiles are detected if not only the same genes are mutated but also if genes located proximally on a network are mutated; this "relaxed" notion of mutation profile similarity leads to improved clustering. (B) Kaplan-Meier survival analysis of clusters from HumanNet-guided clustering of somatic mutation profiles. Each of 14 reported clusters is plotted as a separate survival curve, and the *p*-value of the multivariate log-rank test is displayed. (C) Concordance between different clustering approaches, using ARI. Three of these approaches use the Sample Clustering (sc) pipeline, with HumanNet (hnNet), STRING text mining (sText) or no network (noNet) for guidance. Two

clustering approaches are reproductions from the Hoadley and colleagues ("tcga_mut" obtained from mutation data and "tcga_coca" obtained from multiomics data using COCA). The sixth clustering (disease) is simply a grouping of patients by tumor type. (D) Kaplan-Meier survival analysis of 13 COCA clusters in pan-cancer multiomics data. Users may click the clock icon next to cluster assignments in the Spreadsheet Visualizer to access this display, which uses the current grouping criterion (configurable) for survival analysis. (E) Sample Clustering of pan-cancer multiomics profiles, displayed by the Spreadsheet Visualizer module. Patient profiles are grouped by overall cluster assignment using COCA. The top heat map (blue) shows cluster assignments based on individual omics data types (expr, expression; RPPA, proteomic; CNV, copy number variation; methyl, methylation; miRNA, microRNA). The heat maps below show CNV data for select genes (middle) and mutation data for select genes (bottom) for the same patients. Users can configure the number of rows to display for each data source, the statistical criteria for selecting rows, and their sorting order. The grouping criteria for samples (COCA cluster assignments here) can also be configured. User-selected clinical annotations of patients (primary disease in this view; color bar second from top) may also be displayed. ARI, adjusted rand index; CNV, copy number variation; COCA, cluster of cluster assignment; NBS, network-based stratification; STRING, search tool for recurring instances of neighboring genes.

profiles by Hoadley and colleagues [32] was also knowledge-guided, relying on mutations in similar pathways to group related samples, and survival analysis of their original sample clusters produced similarly significant survival distinction ($p = 4.3 \times 10^{-29}$; Figure H in S2 File). The KnowEnG Sample Clustering pipeline, although producing comparable results in terms of survival distinction among clusters, stands out for its ease-of-use compared with executing the multistep methods of the original analysis. For instance, the user avoids download and harmonization of prior knowledge, installation, and configuration of multiple software, data transformations between steps, and possibly arranging for computing resources capable of compute-intensive steps such as bootstrap sampling (explained below).

Delving deeper into the patient clusters obtained above, we asked whether the clusters recapitulate the tumor types of patients or whether they reveal new structures in the data. To this end, we calculated the adjusted rand index (ARI) [40] between the clusters and tumor types and repeated the process for other approaches to sample clustering, including the multiomics Cluster-Of-Cluster-Assignment (COCA) clustering reported in the work by Hoadley and colleagues [32] (Fig 3C). Interestingly, although there is a high concordance between tumor type and the COCA cluster labels of the work by Hoadley and colleagues [32] (ARI = 0.82), the same is not true for NBS-based clusters from the KnowEnG pipeline (ARI = 0.13) or for the pathway-based clustering of mutation profiles in the original study (ARI = 0.13). In other words, knowledge-guided clustering finds groups of patient mutation profiles that have strong correspondence with survival characteristics yet do not simply track tumor types, suggesting alternative levels of molecular similarity. We explored this possibility in detail (Appendix D in S2 File) and found the clusters to be characterized by mutations in genes from specific and distinct pathways, even when they are mixed in terms of tumor type representation.

**Clustering of multiomics data.** The standard clustering pipeline in KnowEnG may be applied to any type of spreadsheet data to cluster a collection of samples, whereas the knowledge-guided clustering pipeline may be used on any gene-level spreadsheet, in which rows represent genes. We showcase this capability by performing "multiomics clustering" of the same cohort of patients as above. A major advantage of multiomics profiling of patients is that their mutual relationships and hidden group structures revealed by each data type can be consolidated into a more integrative, higher-level clustering that is more informative than any one type of profile alone. This was demonstrated by Hoadley and colleagues [32] through their COCA method. Mimicking their approach, we first clustered the above pan-cancer cohort of patients based on their gene expression, methylation, copy number variation, or protein abundance profiles (Appendix C in S2 File and Table E in S2 Data) separately, using standard clustering. (Knowledge-guided clustering may also be used for all of these profiles except methylation, which is not a gene-level data set). In addition, we considered our knowledge-guided clustering of mutation data reported above and the microRNA clustering from the original publication [32], thus arriving at 6 different ways to partition the cohort into clusters.

Each such clustering assigns a cluster identifier to a patient, and we can thus describe the multiomics profiles of the patient as a succinct "meta-profile" of 6 cluster identifiers. We then used the standard clustering pipeline on these meta-profiles, arriving at 13 clusters (again mimicking the original published analysis [32]) that capture the 6 different omics data sets on the same patients. For this step, we employed the "bootstrap clustering" option of the sample clustering pipeline, which typically yields more robust clustering [41]; the ease of employing this powerful feature is another example of value added by a cloud-based infrastructure. The steps in which different clustering results were combined into common profiles require manipulations with multiple spreadsheets, each being the result of a separate cluster. KnowEnG facilitates these steps, as well as several other common matrix operations, through its "mini pipelines" that are available as notebooks in a Jupyter environment (Appendix E in S2 File).

**Interactive visualization.** Results of the above multiomics cluster analysis were visualized via the Spreadsheet Visualizer module of KnowEnG (Fig 3E), which in addition to displaying multiple spreadsheets as a heat map, allows users to simultaneously visualize various other properties of samples (e.g., cluster assignments provided by COCA, selected clinical annotations such as age, survival months, and primary disease type), offers different ways of sorting, filtering, and grouping the data and provides useful descriptive statistics, such as histograms, in an interactive manner. The interactive visualization also allows us to easily perform survival analysis of the displayed clusters, and we used this feature to find that the new multiomics clusters are strongly concordant with tumor type (ARI = 0.72) and exhibit differences in survival probabilities ($p = 1.0 \times 10^{-150}$; Fig 3D, Appendix F in S2 File) far more prominently than the mutation-only analyses had revealed. The Spreadsheet Visualizer is a powerful data exploration and preliminary analysis tool in its own right (see Appendix G in S1 File for details) and can be utilized independently of the clustering pipeline.

**Clustering for patient stratification.** As an illustration of how the Sample Clustering pipeline may be used for patient subtyping [36], we next clustered breast cancer patients in the METABRIC data set [42] based on genes related to the epithelial to mesenchymal transition (EMT), which is a process involved in metastasis. Following the approach in the work by Emad and colleagues [43], we clustered patients into 2 groups based on the expression of their EMT-related genes (Appendix G in S2 File). Although standard mode of Sample Clustering did not result in clusters with distinct survival probabilities, the knowledge-guided mode achieved significant Kaplan-Meier log-rank $p$-values using either the STRING [44] textmining interaction network ("sText;" $p = 3.1 \times 10^{-4}$) or the HumanNet "hnInt" network ($p = 7.6 \times 10^{-4}$; Figures L and M in S2 File).

## Case Study 2: Gene prioritization for tumor types

A routinely conducted analysis of high-throughput omics profiles is in the determination of genes associated with particular phenotypic conditions or biological processes of interest. Discovery of differentially expressed genes [45] by contrasting transcriptomic profiles before and after treatment or in case versus control experiments, or of genes whose expression correlates with a numeric phenotype such as drug response [46] are prime examples. The Gene Prioritization pipeline in KnowEnG offers this functionality, given a spreadsheet of omics data (genes x samples) and a "phenotype spreadsheet" (phenotypes x samples) that represents one or more phenotypic labels for each sample in the omics spreadsheet. As a simple demonstration of this pipeline, we analyzed expression data from tumor samples in the pancan12 data set introduced above, comparing each tumor type with all others using a $t$ test to identify significant differences in individual gene expression between the groups; this is the standard version of the pipeline (Fig 4A, Appendix A in S3 File).

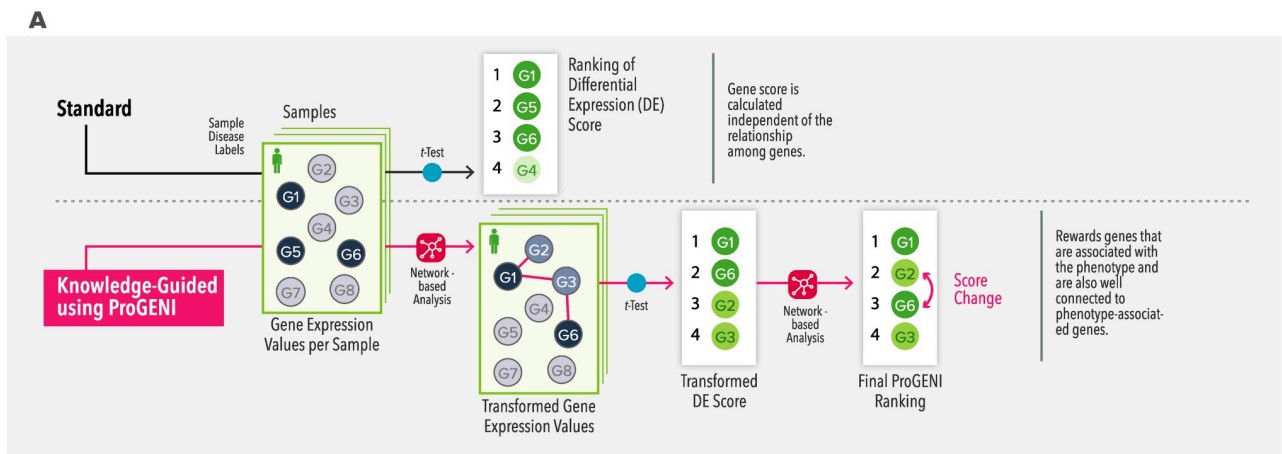

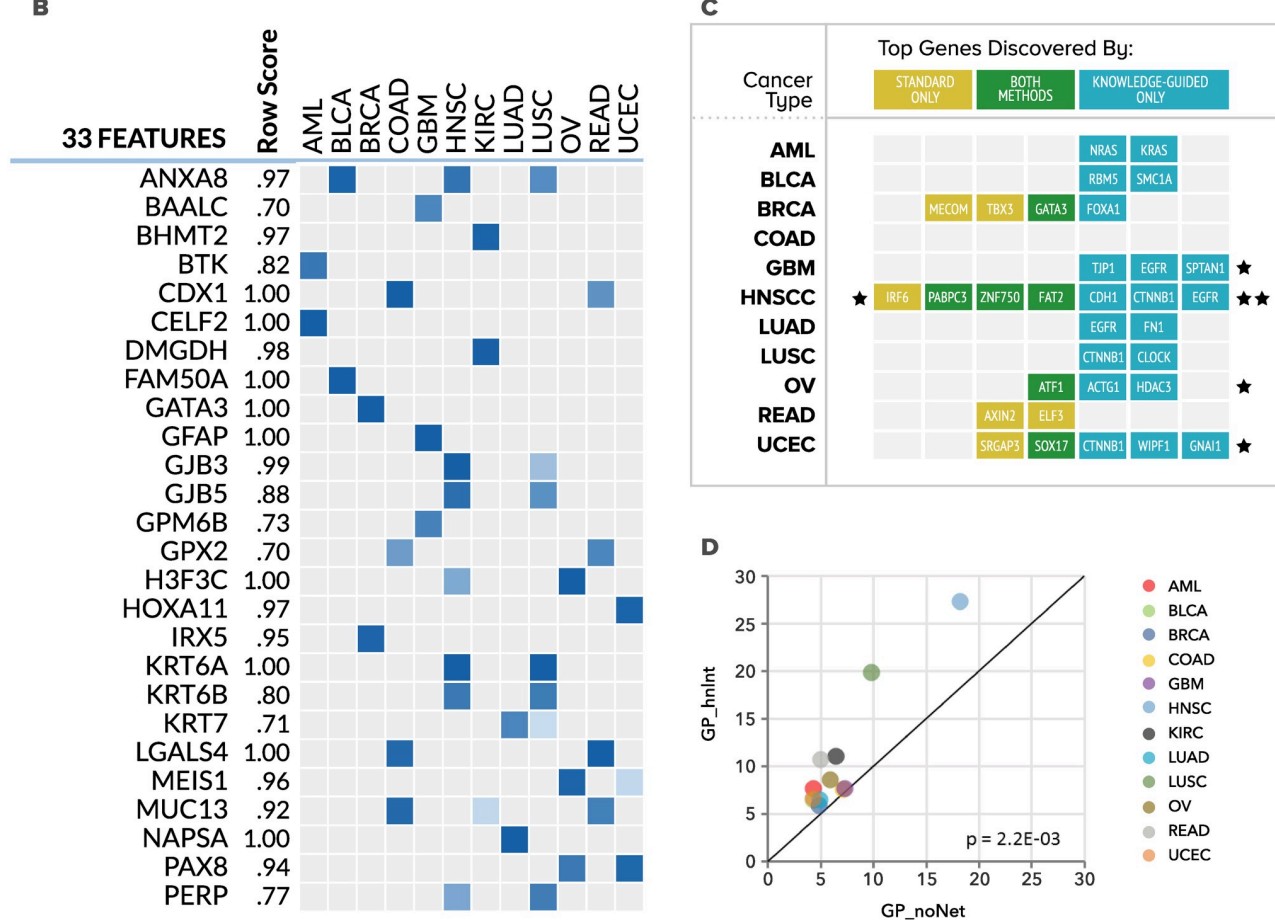

**Fig 4. Knowledge-guided gene prioritization.** (A) In standard mode (top), each gene's expression is tested for association with phenotypic labels, e.g., with a *t* test. In the (bottom) knowledge-guided mode (ProGENI algorithm), each gene's expression is first transformed by taking into account expression levels of its network neighbors, and these "network-smoothed" expression values are tested for association with phenotype. The resulting ranking of genes is subjected to second phase of network-based smoothing to obtain the final ranking. (B) Visualization of results from the Gene Prioritization pipeline, used here to identify top genes associated with each tumor type (based on expression data). Users may choose to analyze and visualize results for multiple phenotypes together and configure how many top genes per phenotype the report should include. (C) Known driver genes for each tumor type that are highly prioritized by standard and/or knowledge-guided modes of Gene Prioritization. (D) Comparison between tumor type–related genes identified using the Gene Prioritization pipeline in standard mode ("GP_noNet") or knowledge-guided mode using HumanNet ("GP_hnInt"), based on their enrichment for GO terms. The axes represent the negative logarithm (base 10) of *p*-value of enrichment between the set of highly prioritized genes (from either method) for a tumor type and the most enriched GO category for that set. GO, gene ontology.

**Knowledge-guided gene prioritization.** KnowEnG also offers a knowledge-guided mode of this pipeline, where the ProGENI algorithm of Emad and colleagues [47] is used to incorporate a network encoding prior knowledge into the identification of phenotype-related genes (Fig 4A), using random walk-based techniques similar to those used in the NBS clustering approach [11]. We had previously tested ProGENI on the task of prioritizing drug response–related genes. Through systematic benchmarking, experimental validations and literature surveys we showed that it identifies phenotype-related genes more accurately compared with simple statistical methods as well as machine learning methods that do not utilize prior knowledge [48]. We now applied this algorithm, via the knowledge-guided gene prioritization pipeline, to identify top genes associated with each tumor type, based on expression data (Fig 4B, Appendix A in S3 File). (KnowEnG allows this analysis to be performed for all tumor types through one simple operation, rather than repeat it for each tumor type separately).

**Gene prioritization finds driver genes.** For an independent assessment of the above results, we compared the top 100 genes for each tumor type with drivers of that cancer as cataloged in the IntOGen database [49] based on mutation and gene fusion data (Fig 4C). We observed overlaps between the 2 lists; for example, in head and neck squamous cell carcinoma (HNSCC), 6 of the highly prioritized genes are known drivers (Fisher's exact test p = $8.2 \times 10^{-4}$; Figure A in S3 File). A similar assessment of genes reported by the standard pipeline (without knowledge-guidance) revealed fewer overlaps with respective driver sets for all but 2 tumor types (Fig 4C). Often, common driver genes were identified by both versions of the pipeline, e.g., GATA3 for breast cancer (BRCA), but in many cases the knowledge-guided version reported known drivers that were missed by the standard pipeline, e.g., FOXA1 for BRCA, NRAS and KRAS for acute myeloid leukemia (AML), and CDH1, CTNNB1, and EGFR for HNSCC. (ESR1, a well known marker of BRCA [50], was ranked in the top 1.2% of all genes for BRCA, but ranked much worse for other tumor types). Similar conclusions were reached when we repeated the assessment using a larger external set of tumor type drivers, based on both IntOGen and COSMIC databases [49, 51] (Appendix A in S3 File).

**Functional enrichment of prioritized genes.** To gain further insights into the highly ranked genes reported for each tumor type in the above analysis, we subjected them to functional enrichment analysis through the Gene Set Characterization pipeline, whose standard version uses the Fisher's exact test to assess the enrichment of a gene set for prespecified annotations. This revealed various interesting pathways and GO terms as being significantly associated with each tumor type (Appendix B in S3 File). For instance, glioblastoma (GBM)-related genes found by ProGENI were significantly associated with receptor proteins in the presynaptic active zone and excitatory synapse, whose altered expression can enhance gliomas ability to grow and survive [52] (Bonferroni corrected $p = 6.0 \times 10^{-3}$). Similarly, AML-related genes were enriched for platelet activation, shown to be related to blast proliferation [53] (Bonferroni corrected $p = 2.0 \times 10^{-6}$). The extent to which significant functional properties can be associated with a gene set extracted by genomics analyses is one measure of the utility of that gene set [54]. Thus, we summarized the results of gene set characterization by noting the most statistically significant functional enrichment (of genes prioritized) for each tumor type. We noted that when the same process was repeated using genes reported by the standard gene prioritization pipeline the functional enrichments tended to be less prominent (Fig 4D), thus providing further evidence of the value of knowledge-guided gene prioritization. The same conclusion was reached when a different network (STRING text mining) was used in gene prioritization instead of the HumanNet integrated network (Appendix B in S3 File).

**Pan-cancer signature from prioritized genes.** Sets of genes of particular relevance to a tumor type are often used as a "signature" of that tumor, i.e., a representative gene set that captures much of the diagnostic or prognostic value of the entire expression profile. The PAM50

signature of breast cancer is a prime example [36], being used for patient stratification based on expression of a small set of genes. We asked if the tumor-associated genes prioritized above for each tumor type together form a similar signature with prognostic value in a pan-cancer context. Indeed, we observed that pan-cancer subtypes obtained from clustering only the expression of the tumor-associated genes were just as predictive of survival (Kaplan-Meier $p = 3.8 \times 10^{-175}$) as the above-mentioned clusters based on entire expression profiles (p = $1.2 \times 10^{-169}$; see Appendix C in S3 File). This finding was robust to the use of different networks (or no network) in the gene prioritization step.

## Case Study 3: Signature analysis and gene set characterization on a third-party system

Our next case study makes use of a fourth pipeline—Signature Analysis (Fig 5A)—to study a transcriptomic data set of ESCC samples [33] and also showcases how KnowEnG tools can be invoked on computing infrastructures external to the platform (Fig 5B). Although the KnowEnG web portal offers a flexible graphical user interface, advanced users performing bioinformatics analysis on a different computing framework may prefer to avail of KnowEnG pipelines on that external framework directly, without tedious transfer of data, intermediate results, or code from one system to another.

**Interoperability.** KnowEnG currently offers such seamless interoperability with the SB-CGC, which provides researchers with secure access to public data sets such as TCGA and TARGET. We used SB-CGC to access RNA-seq data for the previously reported ESCC tumor samples [33] and created a transcriptomic spreadsheet (genes x samples) for further analysis with KnowEnG pipelines in the SB-CGC environment (Fig 5B, Appendix A in S4 File). This is made possible by the publication of KnowEnG pipelines as native workflows on the SB-CGC, with simple graphical interfaces, and creates opportunities for synergistic use of functionalities offered by these 2 powerful genomics computing platforms. (External availability of KnowEnG pipelines includes seamless access to the massive Knowledge Network that supports knowledge-guided analysis). Interoperability is an important tenet of the emerging vision of computing infrastructures of the future. It was achieved by using 2 emerging technologies—Docker containers [5] to make the underlying software of each pipeline portable and Common Workflow Language (CWL) [55] to provide a standardized description of the pipeline (Appendix H in S1 File). This alternative mode of KnowEnG usage also facilitates reproducibility and reusability; e.g., users may share their project on SB-CGC with collaborators. Thus, by ensuring interoperability and reusability, in addition to accessibility and findability already offered by the cloud-based web platform, the KnowEnG-CGC joint framework takes a major step toward the realization of the FAIR principles of modern data science.

**Signature analysis for patient subtyping.** Operating within the SB-CGC framework, we performed a signature analysis of 79 ESCC patients as reported in the original TCGA publication. Signature analysis [56] is a widely used method in cancer informatics and has been used for various tasks such as identifying subtypes [36], characterizing purity of tumor samples [57], determining the abundance of immune cells in tumor microenvironment [58], characterizing transitions involved in the invasion-metastasis cascade [43], etc. Here, given a spreadsheet of transcriptomic profiles of a cohort of patients, and a second spreadsheet of predetermined expression signatures, the pipeline finds the closest matching signature for each patient (Fig 5A). This often allows existing insights about the signature to shed light on clinical characteristics of the patient based on their molecular profile. Following the original publication, we matched ESCC samples to signatures representing 4 subtypes of LUSC [59], because the 2 cancers are anatomically adjacent and previously established subtypes of LUSC

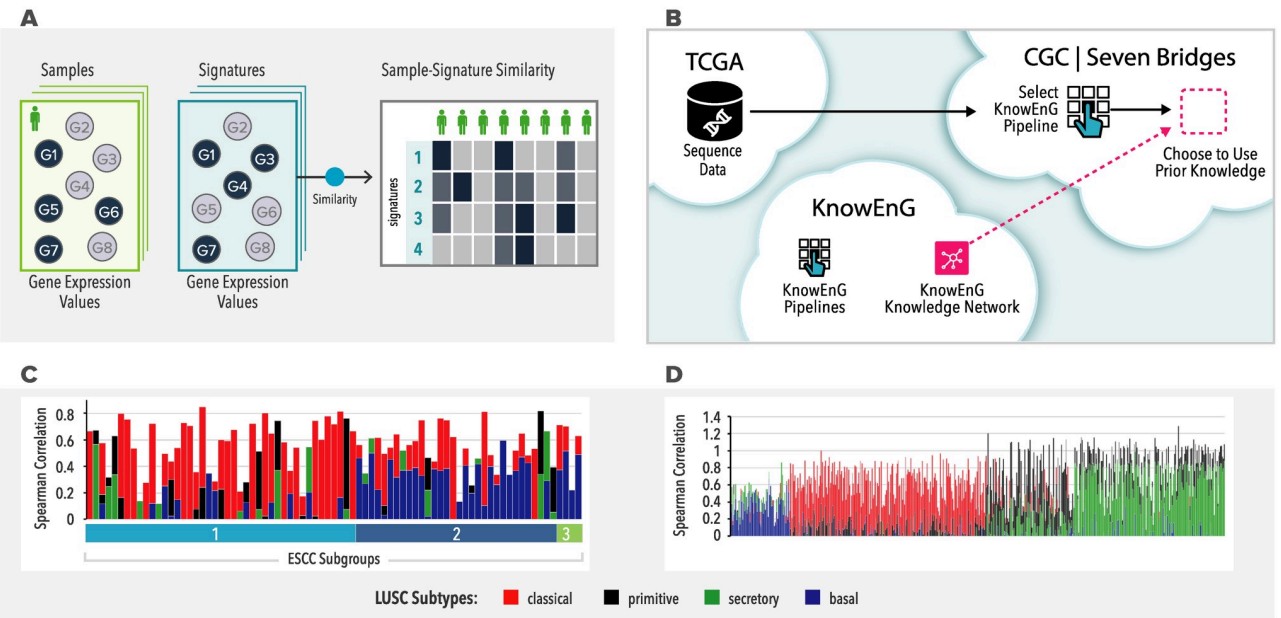

**Fig 5. Signature analysis pipeline.** (A) Each user-uploaded expression profile (sample) is matched against expression profiles in a predetermined collection (signatures) and match scores for all sample-signature pairs are reported by the pipeline. (B) Signature Analysis and other KnowEnG pipelines can be executed seamlessly on the third-party platform of SB-CGC that hosts a large repository of cancer data and associated tools. The pipelines are published on SB-CGC as a native workflow and the Knowledge Network is transferred "under the hood" from the KnowEnG Cloud when needed by a pipeline. (C) Signature Analysis of 79 ESCC samples, distributed into 3 subgroups, matched against 4 LUSC signatures (subtypes) using Spearman's Correlation Coefficient. (D) Signature analysis of 551 LUSC samples available on the SB-CGC, matched against 4 LUSC signatures. ESCC, esophageal squamous cell carcinoma; KnowEnG, Knowledge Engine for Genomics; LUSC, lung squamous cell carcinoma; SB-CGC, Seven Bridges Cancer Genomics Cloud.

may be relevant to ESCC as well (Appendix B in S4 File). We noted that one cluster of ESCC patients ("ESCC1," identified in the original publication) mostly (65%) resembled the classical subtype of LUSC, whereas the second main cluster ("ESCC2") mostly (63%) matched the basal subtype of LUSC (Fig 5C), and fewer samples matched the primitive and secretory subtypes. The correspondence discovered between ab initio detected ESCC subtypes and previously reported LUSC subtypes is generally consistent with the observations of the original TCGA esophageal carcinoma analysis, who note that tumors matching the classical expression subtype also had similar somatic alterations to the subtype and were associated with poor prognosis and chemotherapeutic resistance. To highlight the convenience of co-localizing the analysis workflows with the data on the SB-CGC, we reran the analysis by simply substituting an alternate TCGA data set of LUSC tumor samples, again finding the classical subtype (40%) to be the most prevalent (Fig 5D).

**Pathway analysis of subtype-associated genes.** Having categorized ESCC patients into one of 4 subtypes using signature analysis, we next used the standard gene prioritization pipeline to identify genes associated with each subtype and subjected the resulting subtype-associated gene lists (Appendix C in S4 File) to further analysis using the gene set characterization pipeline introduced above. We now used the knowledge-guided version of this pipeline, which instead of performing the traditional "enrichment test" between sets [60], uses a random-walk algorithm with the user-provided gene set as "restart nodes," to find property nodes of the Knowledge Network that are most related to the given gene set (Fig 6A). This class of algorithms has been successfully used to quantify the relationship between network nodes in a variety of domains such as web mining [61] and social network analysis [62]. The KnowEnG

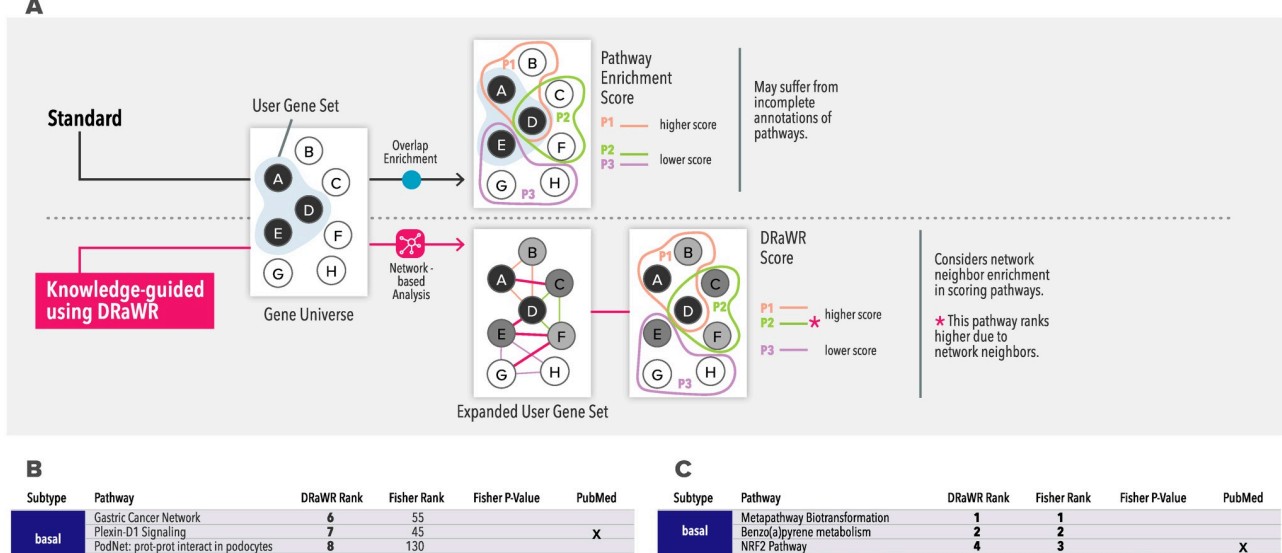

**B**

| Subtype | Pathway | DRaWR Rank | Fisher Rank | Fisher P-Value | PubMed |
|---|---|---|---|---|---|
| basal | Gastric Cancer Network | 6 | 55 | | |
| | Plexin-D1 Signaling | 7 | 45 | | X |
| | PodNet: prot-prot interact in podocytes | 8 | 130 | | |
| | IL17 Signaling Pathway | 9 | 46 | | X |
| classical | TOR Signaling | 9 | 65 | | X |
| primitive | FOXM1 Transcription Factor Network | 6 | 26 | | X |
| | E2F Transcription Factor Network | 7 | 37 | | X |
| secretory | Odorant GPCRs | 1 | 76 | | |
| | p75(NTR)-Mediated Signaling Pathway | 2 | 118 | | X |
| | PPAR Signaling Pathway | 5 | 90 | | X |
| | Calcium Regulation in the Cardiac Cell | 6 | 51 | | |
| | Purin Metabolism | 8 | 75 | | |

**C**

| Subtype | Pathway | DRaWR Rank | Fisher Rank | Fisher P-Value | PubMed |
|---|---|---|---|---|---|
| basal | Metapathway Biotransformation | 1 | 1 | | |
| | Benzo(a)pyrene metabolism | 2 | 2 | | |
| | NRF2 Pathway | 4 | 3 | | X |
| classical | Metapathway biotransformation | 1 | 2 | ★★ | |
| | NRF2 Pathway | 2 | 1 | | X |
| | Glutathione Metabolism | 8 | 3 | | X |
| | Benzo(a)pyrene Metabolism | 5 | 5 | | |
| | Transcriptional Activation by NRF2 | 6 | 6 | | X |
| primitive | Metapathway Biotransformation | 1 | 3 | | |
| | Sphingolipid Metabolism | 2 | 5 | ★ | |
| | Oxidation by Cytochrome P450 | 4 | 6 | ★ | X |
| secretory | Integrins in Angiogenesis | 3 | 1 | ★ | |
| | Syndecan-1-mediated Signaling Events | 4 | 2 | ★★ | X |

**Fig 6. Knowledge-Guided GSC.** (A) Common approaches to GSC examine the overlap (top) between a user-provided gene set (e.g., genes A, D, E) and genes in a pathway (e.g., A, D, B in pathway P1). In the (bottom) knowledge network-guided mode (algorithm DRaWR), the association between 2 gene sets is based not only on direct overlap between them but also on network-based proximity between them. (B) LUSC subtype-associated pathways found exclusively with network-guided GSC pipeline using DRaWR. (C) Pathways associated with LUSC subtypes found by standard as well as network-guided GSC pipelines. GSC, gene set characterization; LUSC, lung squamous cell carcinoma.

pipeline uses an implementation called "DRaWR" [24], the main advantage of which compared to enrichment tests is that it examines not only properties with which the given genes are annotated but also the properties with which genes related to the given genes are annotated (Appendix C in S4 File). We have previously used DRaWR to characterize gene sets in *Drosophila* development [24] and cancer [63]. Here, we used the DRaWR-based knowledge-guided gene set characterization pipeline with the HumanNet Integrated network [39] as the underlying network to identify, for ESCC subtype-related genes, the most related pathways in the Enrichr Pathways Collection [64]. (The pipeline offers several options for the network as well as the properties to be ranked; see Appendix C in S4 File). As a point of contrast, we also analyzed the gene sets with the standard version of the pipeline that uses the traditional Hypergeometric test approach [65]. Fig 6B tabulates 12 discovered pathway associations for ESCC subtypes that were reported by the DRaWR-based version of the pipeline but not by the standard version. Even though these associations do not meet the traditional criterion of significant set overlap, there is support in the literature for 7 of the 12 associations. Moreover, the top-ranked association was between basal subtype of ESCC and the gastric cancer network, which is credible given the close relationship between ESCC and gastric cancer, which are anatomically adjacent and share several risk factors [66]. Surprisingly, this association was not detected by the enrichment test performed in the standard pipeline. Another interesting example is the primitive subtype being linked to FOXM1 transcription factor network but only by the DRaWR-based pipeline. FOXM1 has been found to be related to ESCC progression [67] and to be a potential drug target; our finding of a specific association with the primitive subtype of ESCC suggests that the tumor subtype may be an important factor to consider in its

therapeutic significance. We also found several subtype-pathway associations reported by both versions of the pipeline (Fig 6C). For instance, both the basal and classical subtypes were associated with NRF2 pathway [68], the secretory subtype was linked to Syndecan-1 mediated signaling event [69], and the primitive subtype to oxidation by Cytochromes P450 [70]. A total of 6 of the 13 such associations found by enrichment-based as well as DRaWR-based gene set characterization had circumstantial evidence in the literature.

In summary, this case study illustrates how different KnowEnG pipelines, in this case, beginning with signature analysis and followed by gene prioritization and gene set characterization, can be used in a workflow to not only relate patient profiles to previously reported cancer subtypes but also to glean novel insights about genes and pathways differentiating patients matched to different subtypes. We performed these analyses on a system external to KnowEnG (i.e., SB-CGC), but the same workflow may be executed on the KnowEnG platform as well, and the interface facilitates easy "stringing" of multiple pipelines to enable such workflows.

## Discussion

KnowEnG is an analysis engine designed and implemented with the needs and trends of modern genomics research in mind. KnowEnG offers a vision of genomic computing that is complementary to the dominant paradigm where software packages (e.g., in R or python) are installed on the user's computer and executed locally. The current paradigm is convenient as long as data sets predominantly reside locally, but with the on-going movement toward massive data sets in the public domain [71] and a clear need for moving tools to co-locate with these data, we expect the alternative paradigm embraced by KnowEnG to be increasingly relevant. Its main platform provides a convenient way to analyze the user's uploaded spreadsheets while exploiting massive knowledgebases encoded in the Knowledge Network, and its interoperability with major cloud-based platforms such as SB-CGC showcases the advantages of tools moving to data sources while maintaining the convenient "illusion" of local computation.

### Comparison with existing frameworks

KnowEnG embodies many of the powerful ideas to have emerged in the genomics research over the last decade, including knowledge-guided analysis, cloud-based storage and computing, machine learning and network-mining algorithms, and the FAIR principles for broader impact. KnowEnG draws inspiration from existing analytic tools and systems that have brought the above ideas to the fore. For instance, if we consider noncommercial platforms with web portals that offer multiple analytics functionalities for genomics data sets, some prominent examples that come to mind include Galaxy [72], FireCloud/Terra [73], cBioPortal [74], NCI Genomic Data Commons (GDC) Data Portal [2], GenePattern [75], GeneWeaver [76], GeneMANIA [77], and GenomeSpace [78], among others. Below we clarify the unique position of KnowEnG in the rich milieu of genomics platforms available today.

The Galaxy platform provides convenient wrappers around popular external tools, mainly for next generation sequencing data analysis and many other tasks such as meta-genomics, phylogenetics, and sequence analysis that are outside the scope of KnowEnG. However, Galaxy is not geared toward knowledge-guided analysis of users' omics data spreadsheets, and its "downstream" analysis functions do not include sample clustering, gene prioritization for numeric phenotypic scores, or signature analysis, nor do they include the powerful spreadsheet visualization capability of KnowEnG. FireCloud/Terra similarly allows users to perform a variety of analyses of their own data as well as hosted data sets but does not offer, as KnowEnG does, knowledge-guided analysis or a rich visual interface. The popular cancer-related platforms of cBioPortal and NCI GDC are mainly geared toward visualization and exploration of

omics spreadsheets extracted from the data sets they host rather than users' spreadsheets and also differ significantly from KnowEnG in terms of analytics tasks offered, including knowledge-guided tools. Moreover, FireCloud, cBioPortal, and NCI GDC are entirely focused on cancer data analysis. In contrast, KnowEnG tools are meant for analysis of user-provided omics spreadsheets from any biological domain in any of its 20 currently supported species.

GenePattern is a web-based portal that offers a number of modules for genomics data analysis, similar to KnowEnG pipelines, with similar functions. However, these general-purpose machine learning tools do not fall in the genre of knowledge-guided analysis and do not offer the option of exploiting prior knowledge. On the other hand, GeneWeaver and GeneMANIA are online portals for knowledge-guided analysis of user data but are limited to gene sets, similar to KnowEnG's GSC pipeline, and do not provide tools such as sample clustering and gene prioritization for spreadsheet analysis. Another popular modern platform, GenomeSpace, is a warehouse of diverse virtual workflows that connect several external tools and databases and that have some overlaps with KnowEnG functionalities. However, it does not perform any analysis itself and acts more as a data highway for passing data from one tool to another and is thus dependent on external tools and computing platforms to ensure scalability of its workflows.

We reserve a special mention of the 'geWorkBench' [79], an open-source platform provides standard tools for common tasks such as sample clustering, gene prioritization, and gene set enrichment, comparable to "baseline" options in the corresponding KnowEnG pipelines, as well as specialized tools [80–83] for analyzing a gene expression spreadsheet and a co-expression network simultaneously. However, geWorkBench has to be installed on the user's machine, and its web version has a small subset of tools from its desktop version, reflecting the challenges of offering a centralized web portal to compute-intensive tools with complex inputs. KnowEnG pipelines, on the other hand, embrace the use of computationally intensive but often trivially parallelizable techniques such as bootstrap sampling because of the scalability afforded by cloud computing.

In summary, a careful comparison of the features and goals of some of the major contemporary analytical frameworks reveals that KnowEnG brings complementary capabilities to the user, either in terms of the actual analyses offered, in allowing knowledge-guided analysis, in focusing on user-provided spreadsheet data analysis, in providing a consolidated cloud-based back-end for scalable computation, or in its online interface for visualizing data and results. Importantly, KnowEnG also supports analysis for 20 species currently, making its scope broader than several of the above-mentioned tools that are restricted to human genomic analysis. This is a major feature because knowledge-guided analysis requires integration of prior knowledgebases with tools in a species-specific manner.

## Applications to other biological domains

KnowEnG pipelines analyze spreadsheets of genes (rows) by samples (columns), which tabulate numeric information about each gene in each sample. The information may come from a variety of sources, e.g., high-throughput transcriptomics assays using various technologies, mutation counts at the gene level, copy number variations, etc. The analytical approaches do not make strong assumptions about the source of the data and are, as a result, applicable to any number of biological domains, not just cancer, as we discuss next.

The gene prioritization pipeline may be used in any scenario where a spreadsheet of gene-level measurements (expression levels, mutation counts, copy numbers, epigenomic measurements, etc). is available on a collection of samples, along with a phenotypic score for each sample. For instance, Emad and colleagues [47] used this pipeline to identify genes whose basal

expression in a cancer cell line is predictive of the cell line's response to a cytotoxic treatment. Similar analyses have been performed in other published studies [48], although without incorporating a knowledge network. Other examples of potential applications to gene prioritization for numeric phenotypes include identifying genes whose brain expression levels are predictive of pheromone response in honeybees [84], discovering genes predictive of growth rate in bacteria or yeast [85], and identifying gene families whose size (number of paralogs) in a species is correlated with a numeric score of that species, e.g., eusociality index in bees [86]. Indeed, the potential of this line of analysis is evidenced by the recent publication of a tool specifically for relating expression to traits ("TraitCorr" [87]) as an R package. The task of identifying differentially expressed genes between 2 conditions (binary phenotypes) can also be performed, in a knowledge-guided manner, using the gene prioritization pipeline. The high utility of this task needs no introduction, and many tools are available for it [88]. The unique value of the KnowEnG pipeline is that common statistical tests used for this task, e.g., $t$ test or EdgeR [89], can be combined with "smoothing" of gene expression values based on the Knowledge Network as well as subjected to "bootstrapping" for robustness. (We and others have already demonstrated the value of network-smoothing and bootstrapping in prior work on gene prioritization [47]). These additional features of the pipeline are well supported by a cloud-based platform that offers easy scalability and prestored knowledge networks, thus avoiding the hassles of maintaining compute clusters and downloading large networks for a more traditional "local computation" such as those using Bioconductor packages [90].

Clustering is a pervasive operation in bioinformatics and finds uses in a large number of scenarios. Clustering may be performed within KnowEnG for any of the 20 species supported by it, for any set of experimental conditions, and for any type of omics data that assign numeric measurements to genes, to reveal hidden groupings among the conditions. Whereas the common tools for gene expression clustering focus on the task of grouping genes, the KnowEnG Sample Clustering pipeline is geared toward finding groups of samples/conditions that have similar expression profiles. This distinction is crucial to its use of a knowledge network to guide the clustering, lends it a complementary strength, and is expected to be of increasing utility in the future as the practice of profiling tissue samples from individuals grows more popular [91]. The most common uses of sample clustering are in identifying subgroups in cancer patients, based on transcriptomic as well as other omics data sets, e.g., identifying breast cancer subgroups from copy number variations [92], colon cancer subgroups from gene expression data [93], refinement of breast cancer subtypes based on microRNA expression profiles [94], subtyping of different cancers from somatic mutation data [95], to name a few. Other uses of clustering to group samples include clustering of type 2 diabetes patients as well as obese and healthy subjects to find that type 2 diabetes and obesity have similar expression profiles [96], grouping of brain transcriptomes of honeybee nurses and foragers of different ages to show that each behavioral group has similar profiles [97], clustering of Arabidopsis plants treated with plant activators [98], etc.

The GSC pipeline addresses one of the most commonly performed tasks in genomics analysis, which is sometimes referred to as gene set enrichment analysis (GSEA) and often performed using the GSEA tool [99] or through hypergeometric tests. Studies that use this analytical operation are too numerous to list here, but its popularity is evidenced by the huge following that online tools such as DAVID [60] and Enrichr [64] have. We nevertheless included this pipeline in KnowEnG because it is a natural follow-up for the gene prioritization pipeline, and we expect users to make use of it every time they identify top genes associated with a phenotype. Moreover, the KnowEnG pipeline offers 2 complementary approaches to the above task—the popular approach based on hypergeometric tests (as in DAVID) and a novel approach based on random walks with restart (RWR), which we have published

previously [24] and whose unique value we further demonstrate in Appendix D in S4 File. The RWR-based method not only provides an alternative approach to identifying pathways, GO terms, etc., most relevant to a given set of genes, it does so while accounting for gene-gene relationships encoded in a knowledge network, according unique value to the KnowEnG pipeline. We have also published the use of the RWR-based method to characterize gene sets arising out of a brain transcriptomic study of social behavior in 3 different species [100].

### Flexibility of KnowEnG functionalities

While KnowEnG offers analysis functionalities that have broad applicability, we recognize that there will be many scenarios in which the scientist may require modifications to those functionalities or add components to the available pipelines for maximum benefit. There are a few straightforward steps required for a developer to adding a new tool to the KnowEnG system (detailed in Appendix I in S1 File). All KnowEnG pipelines are published as Docker containers and also have CWL descriptions of their inputs, outputs, and execution steps. Such "packaging" of pipelines not only allows them to be used on any platform as part of the user's established workflows, it also allows the user to add their own custom tools, implemented in R or other languages, into the KnowEnG framework. In some cases, the user may wish to execute a particular KnowEnG pipeline repeatedly, with varying configurations. They may achieve this relatively easily through the containerized versions of KnowEnG tools. Another desired dimension of flexibility in KnowEnG analytics is that of the Knowledge Network. Although we provide several predetermined networks as available options to choose from when running a KnowEnG pipeline, the user may wish to utilize a custom network specialized for their domain of enquiry. This is allowed within the KnowEnG framework, via the Network Prepper tool (Appendix E in S1 File). Flexibility is also supported by the multiple modes of access for KnowEnG tools, including the SB-CGC platform and Jupyter notebooks, which provide programming-savvy users several options for integration with other tools.

Despite our efforts to add provisions for flexible uses of KnowEnG functionalities, we recognize that the system will not be a one size fits all solution for bioinformatics analysis. We expect that most genomics projects will have an "upstream" analysis component in which the data are processed in a more domain- and technology-dependent manner, outside of KnowEnG. Our hope is that such upstream analysis will often result in creation of omics spreadsheets similar to the ones used in our case studies and that their downstream analysis (such as sample clustering, gene prioritization, GSC) can be carried out in the KnowEnG framework in a more universal manner, without the need for unduly cumbersome customizations.

### Material and methods

The details of the data sets and KnowEnG analysis pipelines used in this article are fully described in the supporting information. For each of the 3 case studies, there is a corresponding supporting information file (S2 File, S3 File, and S4 File) that details the methods related to that case study in several appendices. These appendices also include additional interpretations for each analyses as well as all of the nondefault run parameters needed to reproduce the results. Many sections contain links to additional resources in which the actual code, containers, or compute servers can be found. Specific instructions for easily reproducing the main analyses on the KnowEnG platform from the 3 case studies are available in Appendix F in S1 File. S1 File also contains additional information about the design and capabilities of the components of the KnowEnG platform.

## Supporting information

**S1 File. Notes on the KnowEnG system.** Nine appendices of supplemental commentary and additional resources describing the KnowEnG system. KnowEnG, Knowledge Engine for Genomics.
(PDF)

**S2 File. Methods for sample clustering case study.** Seven appendices describing the data, pipelines, methods, and additional analyses that relate to the first case study of performing sample clustering on cancer patient data.
(PDF)

**S3 File. Methods for gene prioritization case study.** Three appendices describing the pipelines, methods, and additional analyses that relate to the second case study of performing gene prioritization with TCGA transcriptomic data. TCGA, the Cancer Genome Atlas.
(PDF)

**S4 File. Methods for signature analysis and GSC case study.** Four appendices describing the data, pipelines, methods, and additional analyses that relate to the third case study of using KnowEnG pipelines to analyze LUSC signatures in ESCC samples on a third-party system. ESCC, esophageal squamous cell carcinoma; KnowEnG, Knowledge Engine for Genomics; LUSC, lung squamous cell carcinoma.
(PDF)

**S1 Data. Tables for KnowEnG system notes.** Eight supplementary tables accompanying S1 File notes on the KnowEnG system. KnowEnG, Knowledge Engine for Genomics.
(XLSX)

**S2 Data. Tables for sample clustering case study.** Thirteen supplementary tables accompanying S2 File appendices related to the sample clustering case study.
(XLSX)

**S3 Data. Tables for gene prioritization case study.** Seventeen supplementary tables accompanying S3 File appendices related to the gene prioritization case study.
(XLSX)

**S4 Data. Tables for signature analysis and GSC case study.** Fourteen supplementary tables accompanying S4 File appendices related to the signature analysis and GSC case study.
(XLSX)

## Acknowledgments

We thank our NIH colleagues, especially Ishwar Chandramouliswaran for his guidance regarding interoperability with Seven Bridges Genomics Cancer Genomics Cloud. We are grateful to the Roy Campbell Systems Research Group, UIUC, and the NIH-BD2K Common Credits pilot program for contributing additional computational resources to perform this study. We also appreciate the assistance and efforts from Seven Bridges Genomics Inc, and from the following UIUC personnel and students: Suyang Chen, Joerg Heintz, Henry Lin, Daniel Meling, Shreya Nagesh, Nathan T. Russell, Noor Shalabi, Jackson W.G. Vaughan, Paul Vijayakumar, Svetlana Vranic-Sowers, and Zhuojun Yao.

## Author Contributions

**Conceptualization:** Charles Blatti, III, Amin Emad, Milt Epstein, Krishna R. Kalari, Liewei Wang, Richard M. Weinshilboum, Jun S. Song, Nahil Sobh, Colleen B. Bushell, Saurabh Sinha.

**Data curation:** Charles Blatti, III, Corey S. Post, Aidan T. Epstein, Saurabh Sinha.

**Methodology:** Charles Blatti, III, Amin Emad, Daniel Lanier, Jinfeng Xiao, Jiawei Han, Nahil Sobh, Saurabh Sinha.

**Project administration:** Subhashini Srinivasan.

**Resources:** Charles Blatti, III, Matthew J. Berry, Milt Epstein, Pramod Rizal, Jing Ge, Xiaoxia Liao, Omar Sobh, Mike Lambert, Peter Groves, Xi Chen, Erik Lehnert, Umberto Ravaioli, Nahil Sobh, Colleen B. Bushell, Saurabh Sinha.

**Supervision:** Amin Emad, Omar Sobh, Richard M. Weinshilboum, Jun S. Song, C. Victor Jongeneel, Jiawei Han, Umberto Ravaioli, Nahil Sobh, Colleen B. Bushell, Saurabh Sinha.

**Visualization:** Amin Emad, Matthew J. Berry, Lisa Gatzke, Milt Epstein, Xiaoxia Liao, Mike Lambert, Peter Groves, Colleen B. Bushell.

**Writing – original draft:** Charles Blatti, III, Amin Emad, Matthew J. Berry, Lisa Gatzke, Milt Epstein, Nahil Sobh, Colleen B. Bushell, Saurabh Sinha.

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
