## [Editor Report · Decision Letter 0]

6 Jun 2019

Dear Dr Blatti, 

Thank you for submitting your manuscript entitled "Knowledge-guided analysis of ‘omics’ data using the KnowEnG cloud platform" for consideration as a Methods and Resources by PLOS Biology.

Your manuscript has now been evaluated by the PLOS Biology editorial staff as well as by an academic editor with relevant expertise and I am writing to let you know that we would like to send your submission out for external peer review.

Please re-submit your manuscript within two working days, ie. by Jun 08 2019 11:59PM.

Kind regards,

Lauren A Richardson, Ph.D

Senior Editor

PLOS Biology

---

## [Decision Letter · Decision Letter 1]

15 Jul 2019

Dear Dr Blatti,

Thank you very much for submitting your manuscript "Knowledge-guided analysis of ‘omics’ data using the KnowEnG cloud platform" for consideration as a Methods and Resources paper at PLOS Biology. Your manuscript has been evaluated by the PLOS Biology editors, an Academic Editor with relevant expertise, and by three independent reviewers.

You'll see that although reviewer #1 is positive about the paper, reviewers #2 and #3 raise significant concerns about the structure and presentation of the manuscript, asking for substantial additional information. In addition, I've included some comments provided by the Academic Editor (see foot of this letter) as this may further guide you in how to address the concerns. I should note that this is not the same person whose advice we sought regarding the initial decision to send your manuscript out for peer review (they were unfortunately not able to continue handling your paper).

Broadly speaking, the concerns seem to revolve around the lack of support for the FAIR credentials of KnowEnG (rev #2, AE), lack of evidence that KnowEnG is substantially distinct from a number of existing systems (rev #2, AE), apparent lack of scope for tailoring of analyses (rev #2), lack of evidence for modularity and future-proofing (rev #3). All of these aspects will need to be clarified and fully supported in order for us consider the manuscript further.

In light of the reviews (below), we will not be able to accept the current version of the manuscript, but we would welcome resubmission of a much-revised version that takes into account the reviewers' comments. We cannot make any decision about publication until we have seen the revised manuscript and your response to the reviewers' comments. Your revised manuscript is also likely to be sent for further evaluation by the reviewers.

Your revisions should address the specific points made by each reviewer and by the Academic Editor. Please submit a file detailing your responses to the editorial requests and a point-by-point response to all of the reviewers' comments that indicates the changes you have made to the manuscript. In addition to a clean copy of the manuscript, please upload a 'track-changes' version of your manuscript that specifies the edits made. This should be uploaded as a "Related" file type. You should also cite any additional relevant literature that has been published since the original submission and mention any additional citations in your response. 

Before you revise your manuscript, please review the following PLOS policy and formatting requirements checklist PDF: http://journals.plos.org/plosbiology/s/file?id=9411/plos-biology-formatting-checklist.pdf. It is helpful if you format your revision according to our requirements - should your paper subsequently be accepted, this will save time at the acceptance stage.

Please note that as a condition of publication PLOS' data policy (http://journals.plos.org/plosbiology/s/data-availability) requires that you make available all data used to draw the conclusions arrived at in your manuscript. If you have not already done so, you must include any data used in your manuscript either in appropriate repositories, within the body of the manuscript, or as supporting information (N.B. this includes any numerical values that were used to generate graphs, histograms etc.). For an example see here: http://www.plosbiology.org/article/info%3Adoi%2F10.1371%2Fjournal.pbio.1001908#s5.

For manuscripts submitted on or after 1st July 2019, we require the original, uncropped and minimally adjusted images supporting all blot and gel results reported in an article's figures or Supporting Information files. We will require these files before a manuscript can be accepted so please prepare them now, if you have not already uploaded them. Please carefully read our guidelines for how to prepare and upload this data: https://journals.plos.org/plosbiology/s/figures#loc-blot-and-gel-reporting-requirements.

Upon resubmission, the editors will assess your revision and if the editors and Academic Editor feel that the revised manuscript remains appropriate for the journal, we will send the manuscript for re-review. We aim to consult the same Academic Editor and reviewers for revised manuscripts but may consult others if needed.

We expect to receive your revised manuscript within two months. Please email us (plosbiology@plos.org) to discuss this if you have any questions or concerns, or would like to request an extension. At this stage, your manuscript remains formally under active consideration at our journal; please notify us by email if you do not wish to submit a revision and instead wish to pursue publication elsewhere, so that we may end consideration of the manuscript at PLOS Biology.

When you are ready to submit a revised version of your manuscript, please go to https://www.editorialmanager.com/pbiology/ and log in as an Author. Click the link labelled 'Submissions Needing Revision' where you will find your submission record. 

Sincerely,

Roli Roberts

Senior Editor

PLOS Biology

REVIEWERS' COMMENTS:

Reviewer #1:

The paper describes KnowEnG computational system for analysis of genomics data sets. The system includes tools for popular omics data analysis tasks such as gene filtering and clustering, sample clustering, gene set analysis etc. In addition to standard tools, the system offers ‘knowledge-guided’ data-mining and machine learning algorithms using their massive Knowledge Network aggregating information about gene relationships from a wide range of resources. In addition to the use of the Knowledge Network, the KnowEnG tools are portable to diverse computing environments, are accessible via web-portal GUI’s and programmatically via Jupyter scripts. In the paper, the power of the system is demonstrated by applying complex sophisticated analytical pipelines in re-analysis of published cancer data sets. Authors also demonstrate interoperability with cloud-based data repositories for seamless access to large datasets.

The KnowEnG system is a significant and very welcome development that has potential of helping with “democratization” of analytical tools accessibility. In the use cases presented in the paper, authors demonstrate the power of the system in performing very complex analytical tasks on large scale multi-omic datasets in a few relatively straightforward steps. While there is always a learning curve in using any system capable of such complex analyses, KnowEnG pipelines provide shortcuts that could result in significant time saving for bioinformatics professionals, and it enables analyses that would otherwise not be accessible to non-specialists. The other potentially game changing aspect of the system is the Knowlege Network that aggregates a massive amount of prior knowledge about gene relationships. Authors describe use of the network by using individual subsets of the prior-knowledge and the ProGENI algorithm to transform input data using the chosen subnetwork, but it is likely that the network itself can become an important test-bed for novel computational algorithms leveraging the whole network. The system is cloud-based and thus offers an alternative to desktop based analysis environments. The advantage of this model has been demonstrated by seamless access to TCGA data sets via Seven Bridges cloud based management system. Altogether, a very nicely written paper describing a significant and potentially highly impactful new analytical platform for analysis of omics data.

Reviewer #2:

There are parts of this paper that I like, such as its focus on FAIR principles and use of modern platforms for implementing its functionality. Overall though, I feel that the benefits of the platform are oversold and/or unsubstantiated, and the focus of the paper is too mixed to have a clear impact.

Specifically, there are two focuses of the paper. The first is that KnowENG is a modern data sciences platform that is applicable to many users, e.g. as the authors state it marks "a significant step towards realizing the ‘FAIR’ vision". I don't find the support in the paper for this statement particularly compelling. There is no comparison to other extremely widely used platforms such as Galaxy, GenomeSpace, Terra, or DNANexus (some are not even mentioned). There is no discussion of how many users the platform has. Statements like "its tools being applicable to any data set 105 comprising gene-level measurements or scores for a collection of samples" are abstract and do not spell out why the tools are more valuable than those already available through other online platforms. Instead, there are three examples focused on cancer analysis, with only a reassurance that "The scope of KnowEnG analytics goes far beyond cancer analysis". It is possible that KnowENG does represent a significant leap as a general platform, but the manuscript does not support this claim adequately. Instead, it reads like a platform the authors developed informed by their research needs and now claim as relevant to other users without adequate evidence for this claim.

The second focus is that users can use KnowENG to reveal insights into their data. Perhaps this is my bias, but if I were to analyze an 'omics dataset I had generated, I would likely have a hypothesis in mind and care deeply about the underlying "knowledge graph" that was used to support clustering/prioritization/etc. For example, I may want to use a graph that only includes datasets in tissues relevant to my disease, or to exclude certain datasets I may not trust. Without the ability to tailor data/analysis to my hypothesis, the described workflows will only yield broad insights; e.g. "A log-rank test revealed highly significant distinction across the clusters in terms of survival probabilities (p-value 3.7E-33)". This is nice but leaves me unclear on what the next steps would be (are the results of a KnowENG analysis publishable without anything further? do they suggest an experiment?). Lines 165-174 seem to support this view:

165 Interestingly,

166 while there is a high concordance between tumor type and the COCA cluster labels of Hoadley

167 et al.[4] (ARI = 0.82), the same is not true for NBS-based clusters from the KnowEnG pipeline

168 (ARI = 0.13) or for the pathway-based clustering of mutation profiles in the original study (ARI =

169 0.13). In other words, knowledge-guided clustering finds groups of patient mutation profiles that

170 have strong correspondence with survival characteristics yet do not simply track tumor types,

171 suggesting alternative levels of molecular similarity. We explored this possibility in detail

172 (Supplementary Note SN7), and found the clusters to be characterized by mutations in genes

173 from specific and distinct pathways, even when they are mixed in terms of tumor type

174 representation

It is not surprising to me that these clusters would not match tumor types, since the KnowENG analysis was not designed to find tumor types (instead, its reliance on large-scale databases makes it unsurprising it clusters according to genes/pathways, since most databases will capture pathway information). What if I (as in Hoadley et al) wanted to cluster by tumor type? Likely I would need to take much more care in defining my underlying analysis and prior data; can I do that via KnowENG? What if I wanted to understand whether the KnowENG clustering was driven by one dataset, or "promiscuous" genes across the network that had a high weight in the analysis? Can I use KnowENG to conduct follow-up analyses?

It is possible that KnowENG in fact addresses my comments above. But the manuscript as written does not make it clear that it does. The manuscript should either (a) focus on the value of a new FAIR platform, by comparing KnowENG to existing platforms (see above), supporting that many users have found it to be of value, and being more clear on the value added by its tools; or (b) focus on KnowENG as enabling researchers to conduct specific analyses/workflows, by giving evidence that it yields publishable insights, or at minimum suggests follow-up experiments/analyses.

Reviewer #3:

Interpretation of the deluge of biomedical data, wether publicly available or user-generated, is at the basis for the majority of current research.

Tools like KnowEng represent the link between researchers with a strictly “wet” background” and the multitude of tools that allow meaningful interpretation.

We like the integrative approach of KnowEng and the fact that it is based on current and sensible software principles (i.e. interoperability, web-based access, etc).

If we were to evaluate this manuscript in terms of the concept it implements, our concerns would only be the fact that most bioinformatics analysis nowadays are made in the R framework, and we are not sure how this fits in the KnowEng framework. We routinely create custom scripts to perform very specific analysis steps, and if KnowEng does not allow for those to be integrated, it would affect the flexibility of the framework. 

However, our main concern is related to the software engineering part of the software. 

For example, we looked at the feature prioritization pipeline. This step is, in our opinion, one of the most crucial parts of every pipeline that makes use of high-throughput data. For example, in transcriptomic experiments, when all (or most) genes/transcripts are quantified, the presence of noise is inevitable, and feature prioritization is absolutely mandatory for removing such noise and extract meaningful variables.

KnowEng implements two very simple and outdated methods for the prioritization of important features. We do understand that these are very general methods and as such they will be appropriate (in that they do not make too many assumptions on the distribution of the data) for most datasets, but today, these would cut out the majority of the existing data. In particular, both Pearson correlation and Student’s t-test would be inappropriate for RNA-Seq data due to the particular distribution of count data (Negative binomial), and it seems that KnowEng does not allow for using appropriate tools like EdgeR or DeSeq. Even for microarray data we would never use such measures, instead preferring moderated t-test (as in the Limma package, see Smith et al 2005) or SAM (Tusher et al, 2001). 

This only points to the main shortcoming of all bioinformatics pipeline frameworks: since it is impossibile to allow for all state of the art tools to be present in a single framework, the tool MUST be flexible enough for users to be able to easily plug in new blocks of the pipeline as needed. This ability needs to follow well established software engineering principles.

If KnowEng supports such flexibility, this is never explained in the text nor in the supplementary information. We understand that PlosBio could be seen as more “bio” oriented, but it is a software tool we are talking about, and as such the software part has to be described in detail and justified. The results presented in the manuscript are great, but they represent a small set of datasets analyzed.

In order to accept this manuscript, we would need these aspects to be present and clear in the text.

COMMENTS FROM THE ACADEMIC EDITOR:

I have been through the work, read the reviews and also had a look at the website. I too have major reservations about this paper. On the positive side, there has clearly been a lot of time and effort put into this platform and they have gone to considerable lengths to support users with the provision of tutorials etc. I too like the emphasis on cloud compute and FAIR principles, even if as reviewer 2 states they are invoked but it is not necessarily spelt out how they conform to them. The user interfaces also look pretty clean and useable/useful and KnowEnG may indeed be a useful new platform.

On the negative side I share many of the reservations of reviewers 2, 3. I also did not like the way the resource was presented. Each functionality is described and then applied to a different ‘use case’ dataset and in each instance we were told that KnowEnG outperformed standard methods. Ultimately I found this a pretty unsatisfactory way to present the functionality of the tool as the reader is not really given enough information about each dataset, the insights generated are provided out of context and I found these sections to just provide a superficial overview of a dataset I was not invested in and therefore it ended up being a dull read. Most the methods behind this tool are published we are told, so why do we need to see them applied and justified here? For me a more descriptive work would have been better or even a protocols paper. If there is a new resource such as this my main interest in reading about it are, how do I access it, what data do I need to feed in from which analysis platforms, what analysis routines can I run on the data and how do I do this, what form do the results come back to me in and what advantages does it offer relative to other tools/platforms. At the moment, I don’t get this from the paper in its current form.

---

## [Decision Letter · Decision Letter 2]

20 Nov 2019

Dear Dr Blatti,

Thank you for submitting your revised Methods and Resources entitled "Knowledge-guided analysis of ‘omics’ data using the KnowEnG cloud platform" for publication in PLOS Biology. I have now obtained advice from two of the original reviewers and have discussed their comments with the Academic Editor. 

While reviewer #2 remains to be fully convinced, the Academic Editor is persuaded by your responses to her/his own comments, and by your revisions overall, that we should probably accept your manuscript for publication, assuming that you will modify the manuscript to address the remaining points raised by the reviewers. Please also make sure to address the data and other policy-related requests noted at the end of this email.

IMPORTANT: The Academic Editor feels that your manuscript is still lacking in clarity, and that its length may be precluding a clear impression of the advance. Specifically, s/he says "My inclination is that you accept on the proviso that they look at what they have, which has probably been expanded considerably in the trying to address all our comments and seek to reduce the work, not because they have to, as you say there is no limit on M&R papers but because it would be better vehicle to advertise their resource - which is clearly what they are trying very hard to achieve." The Academic Editor is essentially pointing out that while shortening the manuscript is not required to meet any format requirements for the journal, it may be in your interests (in order to maximise appeal and uptake from the readers) to take a long cold look at your paper with the eyes of an outsider, and see whether you can prune it into a more accessible shape.

We expect to receive your revised manuscript within two weeks. Your revisions should address the specific points made by each reviewer. In addition to the remaining revisions and before we will be able to formally accept your manuscript and consider it "in press", we also need to ensure that your article conforms to our guidelines. A member of our team will be in touch shortly with a set of requests. As we can't proceed until these requirements are met, your swift response will help prevent delays to publication.

*Copyediting*

*Published Peer Review History*

*Early Version*

*Submitting Your Revision*

Sincerely,

Roli Roberts

Senior Editor

PLOS Biology

DATA POLICY:

I should say that it seems that all of the main Figs and most of the Supp Figs are either schematics or screenshots/outputs from the KnowEng platform itself (and so depend on data that can be accessed and manipulated through that portal). However, there are a few Supp Figs which look like they have a simpler structure, and may be presenting data that are not directly output from KnowEng. If this is the case, regardless of the method selected, please ensure that you provide the individual numerical values that underlie the summary data displayed in those figure panels as they are essential for readers to assess your analysis and to reproduce it. NOTE: the numerical data provided should include all replicates AND the way in which the plotted mean and errors were derived (it should not present only the mean/average values).

REVIEWERS' COMMENTS:

Reviewer #2:

I appreciate the authors response to my comments. In particular, the description of how KnowEng fits in with other platforms is very useful, and I think significantly improves the paper.

I think it is clear that KnowEng is distinct from these other platforms. But it seems different mostly in degree, not in kind, and it is still not clear how big of a need it fills. "Knowledge-guided analysis" is not something I have heard used before, and I am not sure how big of a field it is (I could not tell if this is a new term defined to describe KnowEng, or if it was a pre-existing gap for a lot of users).

The example use cases are fine. But they are too detailed for somebody outside of the field of cancer genomic to really grasp the big picture. On the other hand, the high-level descriptions of "clustering" or "gene prioritization" are so broad that it's hard to see what is provided by KnowEng. It seems like the right level of detail would be somewhere in between, where a broad need were motivated and then addressed. I think the long responses to my comments illustrate the challenges in explaining the value of a system like this to somebody who has not developed it or used it closely.

In short, while I see the potential value and novelty of the platform, the current manuscript does not make it clear to me that it is the major advance that the abstract claims.

Reviewer #3:

Some of my concerns are still there, in particular the fact that the tool requires the maintainer to put an effort in including new analysis approaches. However, this is true for most available pipeline managers like KnowEng. 

Most of my other comments have been addressed appropriately, and the amount of work that was put in the revision gives me hope that the authors will keep maintaining and developing KnowEng into a more complete framework.

---

## [Editor Report · Decision Letter 3]

19 Dec 2019

Dear Dr Blatti III,

On behalf of my colleagues and the Academic Editor, Thomas C. Freeman, I am pleased to inform you that we will be delighted to publish your Methods and Resources in PLOS Biology. 

Early Version

PRESS 

Kind regards,

Sofia Vickers

Senior Publications Assistant

PLOS Biology

On behalf of, 

Roland Roberts,

Senior Editor

PLOS Biology